# The Combination of ATM and Chk1 Inhibitors Induces Synthetic Lethality in Colorectal Cancer Cells

**DOI:** 10.3390/cancers15030735

**Published:** 2023-01-25

**Authors:** Yuri Tozaki, Hiromasa Aoki, Rina Kato, Kohki Toriuchi, Saki Arame, Yasumichi Inoue, Hidetoshi Hayashi, Eiji Kubota, Hiromi Kataoka, Mineyoshi Aoyama

**Affiliations:** 1Department of Pathobiology, Nagoya City University Graduate School of Pharmaceutical Sciences, 3-1 Tanabe-dori, Mizuho-ku, Nagoya 467-8603, Japan; 2Department of Cell Signaling, Nagoya City University Graduate School of Pharmaceutical Sciences, 3-1 Tanabe-dori, Mizuho-ku, Nagoya 467-8603, Japan; 3Department of Innovative Therapeutic Sciences, Cooperative Major in Nanopharmaceutical Sciences, Nagoya City University Graduate School of Pharmaceutical Sciences, 3-1 Tanabe-dori, Mizuho-ku, Nagoya 467-8603, Japan; 4Department of Gastroenterology and Metabolism, Nagoya City University Graduate School of Medical Sciences, 1 Kawasumi, Mizuho-cho, Mizuho-ku, Nagoya 467-8601, Japan

**Keywords:** ATM, Chk1, CDK1, cancer therapies, cell cycle checkpoint, synthetic lethality

## Abstract

**Simple Summary:**

DNA damage response (DDR)-related proteins contribute to tumorigenesis, tumor progression, and drug resistance. Pharmacological inhibition of DDR-related proteins, such as checkpoint kinase 1 (Chk1), exerts antitumor effects but has serious side effects. Therefore, we attempted to specifically kill cancer cells with low concentrations of drugs by simultaneously inhibiting two DDR-related proteins. In this study, we demonstrated that the combined treatment with ataxia telangiectasia-mutated serine/threonine kinase (ATM) inhibitor (ATMi) and Chk1 inhibitor (Chk1i) exerts synergistic antitumor effects and induces cancer-specific synthetic lethality at low doses. The ATMi and Chk1i combination synergistically promotes CDK1 activation, resulting in the induction of apoptosis with enhanced cell cycle progression in cancer cells. Considering that the combined treatment exerts antitumor effects at one-tenth the concentrations described in previous reports, with high antitumor efficacy and few side effects it could be an effective treatment approach.

**Abstract:**

Genetic abnormalities induce the DNA damage response (DDR), which enables DNA repair at cell cycle checkpoints. Although the DDR is thought to function in preventing the onset and progression of cancer, DDR-related proteins are also thought to contribute to tumorigenesis, tumor progression, and drug resistance by preventing irreparable genomic abnormalities from inducing cell death. In the present study, the combination of ataxia telangiectasia-mutated serine/threonine kinase (ATM) and checkpoint kinase 1 (Chk1) inhibition exhibited synergistic antitumor effects and induced synergistic lethality in colorectal cancer cells at a low dose. The ATM and Chk1 inhibitors synergistically promoted the activation of cyclin-dependent kinase 1 by decreasing the phosphorylation levels of T14 and Y15. Furthermore, the combined treatment increased the number of sub-G1-stage cells, phospho-histone H2A.X-positive cells, and TdT-mediated dUTP nick-end labeling-positive cells among colon cancer cells, suggesting that the therapy induces apoptosis. Finally, the combined treatment exhibited a robust antitumor activity in syngeneic tumor model mice. These findings should contribute to the development of new treatments for colorectal cancer that directly exploit the genomic instability of cancer cells.

## 1. Introduction

Colorectal cancer is one of the most frequently diagnosed cancers and the second leading cause of cancer death worldwide [1]. Advances in understanding of the pathophysiology of colorectal cancer have led to the development of various therapies, including cytotoxic agents, molecular-target agents, and immunotherapies, but the efficacy of many of these therapies remains unsatisfactory [2]. Thus, more-effective approaches to therapeutic intervention for patients with colorectal cancer need to be developed.

The cell cycle checkpoint is a regulatory mechanism that ensures cells progress through the cell cycle correctly in the G1/S, S, and G2/M phases [3]. Genetic abnormalities, such as DNA damage, induce the DNA damage response (DDR) which mediates DNA repair at cell cycle checkpoints. The genomic integrity of cells is often threatened by DNA damage caused by factors such as irradiation and UV light as well as metabolic and replicative stress [4]. The DDR detects damage and protects cells from genotoxicity by regulating the cell cycle and repairing DNA [2]. However, if severe or irreparable DNA damage occurs, the DDR induces apoptosis or cellular senescence to prevent the transmission of genetic lesions to daughter cells [4]. The DDR is regulated by two major signaling pathways, the ataxia telangiectasia-mutated serine/threonine kinase (ATM)-checkpoint kinase 2 (Chk2) pathway in response to DNA double-strand breaks (DSBs), and the ataxia telangiectasia-mutated and Rad3-related serine/threonine kinase (ATR)-checkpoint kinase 1 (Chk1) pathway in response to DNA single-strand breaks (SSBs), and these pathways are partially complementary [5]. Chk1 and Chk2 regulate the activity of target proteins, including p53, CDC25, and Wee1, via phosphorylation [5]. Cyclin-dependent kinase 1 (CDK1), which functions downstream of the DDR, is a master regulator of cell cycle progression. Along with cyclin B, CDK1 up-regulates the expression of proteins required for mitotic M phase initiation and promotes G2/M transition. In the G2 phase of normal cell cycle progression, CDC25 activates CDK1 by the dephosphorylation of T14 and Y15 [6]. In contrast, induction of the DDR by DNA damage promotes the phosphorylation of CDK1 T14 and Y15, which inactivates CDK1 through various pathways, resulting in cell cycle arrest until DNA repair is completed [6].

Although the DDR is thought to play a role in preventing cancer onset and progression, DDR-related proteins have been associated with greater predisposition to cancer development, progression, and drug resistance [7,8]. Thus, DDR-related proteins are thought to be therapeutic targets in various cancers [9,10]. Overexpression of Chk1 has been reported in many types of cancer and linked to malignant phenotypes [11]. Various small-molecule Chk1 inhibitor (Chk1i) drugs have been developed, some of which have reached clinical trials in combination with chemotherapies [12]. ATM is also thought to be involved in cancer progression, metastasis, and chemoresistance [13]. Thus, ATM-targeting drugs hold promise in treating cancers, and one such drug has reached clinical trials [14]. In cancer cells, the ATM-Chk2 pathway is often attenuated; thus, cancer cell survival often depends on the ATR-Chk1 pathway [15,16,17,18,19]. Although some Chk1i drugs have entered clinical trials due to their promising efficacy, no agent within this class of kinase inhibitors has reached phase III evaluation or received FDA approval [20]. For instance, the clinical trial of the Chk1i AZD7762 was terminated due to multiple adverse effects, including cardiac toxicity [21]. In addition, past clinical trials suggest that combining a Chk1i with traditional cytotoxic DNA-damaging agents induces normal tissue toxicities that outweigh the minimal gains in therapeutic efficacy [20]. These findings highlight the need for therapeutic interventions that act on cancer cells at lower concentrations.

“Synthetic lethality” is defined as a type of genetic interaction in which the combination of more than two genetic events results in cell death. Synthetic lethality is thought to kill cancer cells specifically without affecting normal cells by acting on specific genes or common molecular pathways regulated in the carcinogenesis process. For example, synthetic lethality can be induced by poly(ADP-ribose) polymerase 1 inhibitors in breast and ovarian cancer cells with inactivating mutations in the BRCA1/2 genes [22]. Another example of the selective killing of cancer cells is the treatment of non-small cell lung cancer cells with driver mutations in the Kirsten rat sarcoma viral oncogene homologue (K-ras) with tumor necrosis factor-related apoptosis-inducing ligand and a small molecule mimic of the second mitochondria-derived activator of caspase [23]. Synthetic lethality has been applied to therapies that target the DDR and holds great promise as a cancer-selective treatment [24]. Numerous reports have described the effectiveness of checkpoint-related molecules and inhibitors in cells with genetic mutations, as well as the effects of other chemotherapeutic agents and radiation therapy [25,26,27,28]. We hypothesized that the efficacy of Chk1is could be enhanced by inhibiting the ATM-Chk2 pathway, which plays a role complementary with the ATR-Chk1 pathway in the DDR. Specifically, we speculated that simultaneous inhibition of the SSB and DSB repair mechanisms would prevent cancer cells with genomic instabilities from repairing aberrant DNA via the DDR, thereby triggering synthetic lethality. In the present study, we examined the efficacy of the combined treatment of the ATM inhibitor (ATMi) KU-60019 and Chk1i LY2606368 against cultured colorectal cancer cells and tumors in syngeneic tumor model mice.

## 2. Materials and Methods

### 2.1. Cell Culture and Drug Stimulation

HCT116, HT29 human colorectal cancer cells, CT26 mouse colon cancer cells, and SF-TY human normal fibroblasts were purchased from the American Type Culture Collection (ATCC, Manassas, VA, USA) and cultured in low-glucose (1000 mg/dL) Dulbecco’s modified Eagle’s medium (Wako Pure Chemical, Osaka, Japan) supplemented with 10% fetal bovine serum (FBS), 100 U/mL penicillin, and 100 μg/mL streptomycin. DLD-1 human colorectal cancer cells were purchased from the ATCC and cultured in RPMI-1640 (Wako Pure Chemical) supplemented with 10% FBS, 100 U/mL penicillin, and 100 μg/mL streptomycin. Cells were cultured at 37 °C in a 5% CO_2_, 95% air environment. The ATMi (KU-60019) (AdooQ BioScience, Irvine, CA, USA) and Chk1i (LY2606368) (AdooQ BioScience) were dissolved in dimethyl sulfoxide (DMSO) to prepare 10 mM stock solutions. The stock solutions were further diluted in medium and added to the culture medium during stimulation.

### 2.2. Cell Viability Assay

Cell viability was determined using a WST-8 cell proliferation assay (Cell Counting Kit-8, Dojindo Laboratories, Kumamoto, Japan) in accordance with the manufacturer’s instructions. HCT116, HT29, DLD-1, and SF-TY cells dissociated with 0.25% (*w*/*v*) trypsin/1 mM EDTA-4Na solution (Wako Pure Chemical) were seeded into 96-well plates (Corning, Corning, NY, USA) at 5.0 × 10^3^ cells/well. Cells were incubated for 48 h after drug stimulation and then incubated with WST-8 assay reagent for 1 h, after which the absorbance at 450 nm was measured using an iMark^TM^ Microplate Absorbance Reader (Bio-Rad, Hercules, CA, USA). Plotted data were fitted to a four-parameter logistic curve (using ImageJ function, curve fitting, ver. 1.8.0; https://imagej.nih.gov/ij/download.html; date of last access: 1 June 2020; National Institutes of Health, Bethesda, MD, USA), and IC_50_ values for the inhibitors were calculated. The combination index (CI) was used to evaluate the effectiveness of the combined ATMi and Chk1i treatment. Based on the results of the WST-8 cell proliferation assays, the concentration at which effect X was obtained for each drug as a single agent and the concentration at which effect X was obtained in combination were determined using the median effect method. CIs were calculated by assignment [12], where a CI < 1 was evaluated as synergistic, CI = 1 as additive, and CI > 1 as antagonistic.

### 2.3. Western Blotting Analyses

HCT116, HT29, and DLD-1 cells were stimulated for 6 h in four groups: control group, ATMi 10 µM group, Chk1i 0.03 µM group, and combination group. Cells were then washed with PBS, 1× sodium dodecyl sulfate (SDS) sample buffer (62.5 mM Tris-HCl [pH 6.8], 2% SDS, 10% glycerol, 0.02% bromophenol blue, 5% β-mercaptoethanol) was added, and the cells were collected using a scraper. After reducing the viscosity by sonication, the samples were boiled at 95 °C for 3 min and subjected to SDS–polyacrylamide gel electrophoresis. The resolved proteins were electroblotted onto a membrane at 15 V for 40 min, and the membrane was immersed in a 5% skimmed milk solution and shaken at room temperature for 1 h. After washing with TBS containing 0.1% Tween 20 (TBS-T), the membrane was incubated with primary antibody and shaken overnight at 4 °C. The next day, the membrane was washed with TBS-T and incubated with the appropriate secondary antibody conjugated with horseradish peroxidase (Cell Signaling Technology, Danvers, MA, USA; catalog numbers 7074 and 7076) diluted 10,000-fold in TBS-T with shaking at room temperature for 1 h. After washing with TBS-T, protein bands were visualized using an Amersham Imager 600 system (GE Healthcare Life Sciences, Chicago, IL, USA). The antibodies used in Western blotting analyses are listed in Appendix A (diluted in TBS-T supplemented with 3% bovine serum albumin (BSA) and 0.1% NaN_3_).

### 2.4. Cell Cycle Analysis

After stimulation for 48 h, HCT116 and SF-TY cells in the four groups (control, ATMi, Chk1i, and combination) were dissociated with TrypLE™ Express Enzyme (1×) without phenol red (Gibco, Gaithersburg, MD, USA), and then 1 × 10^6^ cells were transferred to tubes and centrifuged. The resulting cell pellets were fixed using ice-cold 70% ethanol and kept at −20 °C overnight. The next day, the fixed cells were centrifuged and washed with PBS, after which FxCycle™ PI/RNase staining solution (Invitrogen, Carlsbad, CA, USA) was added to the cell pellets and left to stand under light-shielded conditions for 30 min. The cells were then analyzed using a FACS Verse™ system (BD Biosciences, San Jose, CA, USA).

### 2.5. Immunofluorescence Staining

HCT116 and SF-TY cells were seeded into CellCarrier-96 plates (PerkinElmer) at 5 × 10^3^ cells/well, and after 24 h of incubation, the cells were divided into four groups (control, ATMi, Chk1i, and combined treatment) and cultured for 48 h. Cells were fixed for 30 min at room temperature in 4% paraformaldehyde, washed with PBS, and then permeabilized for 5 min in PBS containing 0.2% Triton X-100. After washing with PBS, cells were blocked with PBS containing 3% BSA, 0.1% glycine, and 0.1% NaN_3_ for 1 h and then incubated overnight with the primary antibodies, anti-phospho-histone H2A.X (Ser139) (catalog number 9718; Cell Signaling Technology) and anti-53BP1 (catalog number 4937; Cell Signaling Technology). After washing with PBS, the cells were incubated with secondary antibody (catalog number A11070; Invitrogen) and 1 μg/mL of 4,6-diamidino-2-phenylindole (DAPI; DOJINDO) for 1 h at room temperature. An Operetta high-content imaging system (PerkinElmer, Waltham, MA, USA) for phospho-histone H2A.X and an LSM800 confocal microscope (Carl Zeiss, Oberkochen, Germany) for 53BP1 foci were used to observe the stained samples. Harmony^®^ high-content analysis software (Harmony software) (PerkinElmer) was used to determine the number of positive cells. The antibodies used in immunofluorescence staining were diluted in PBS supplemented with 0.2% Triton X-100.

### 2.6. TdT-Mediated dUTP Nick-End Labeling (TUNEL)

TUNEL was performed using an in situ apoptosis detection kit (TaKaRa Bio Inc., Otsu, Japan). After 24 h of incubation, the cells were divided into four groups (control, ATMi, Chk1i, and combined treatment) and cultured for 24 h. The cells were then fixed for 30 min at room temperature in 4% paraformaldehyde, washed with PBS, and then permeabilized for 5 min in PBS containing 0.2% Triton X-100. Next, 30 µL of ice-cold reaction preparation (3 µL of TdT enzyme + 27 µL of Labeling Safe Buffer) was dropped onto cells on Parafilm and incubated for 90 min in a humidified chamber. After washing with PBS, cells were placed on a glass slide with encapsulant (ProLongTM Diamond anti-fade mountant with DAPI) and overlaid with a cover glass. The slides were left in the dark at room temperature overnight and then observed using an LSM800 confocal microscope.

### 2.7. In Vivo Experiment

The present study was approved by the Animal Care and Use Committee of Nagoya City University Graduate School of Pharmaceutical Sciences. All experiments were performed in accordance with the institutional and U.S. National Institutes of Health guidelines for the care and use of laboratory animals. BALB/c mice (female, 5–6 weeks old) received subcutaneous implants of 1.0 × 10^4^ CT26 cells/µL, and tumor engraftment was confirmed on day 7. Intraperitoneal drug administration was started on day 10, and tumor size (long diameter × short diameter × short diameter × 1/2) was measured twice per week until day 31. All drugs were dissolved in DMSO. The dosing regimen was based on previous reports; KU60019 was administered once per day for five consecutive days during the first week and then not administered [29]; LY2606368 was administered two times on one day each week for the first three weeks [28].

### 2.8. Statistical Analysis

All statistical analyses were performed using EZR (Saitama Medical Center, Jichi Medical University, Saitama, Japan), a graphical user interface for R. Specifically, EZR is a modified version of R Commander designed to add functions frequently used in biostatistics [30]. One-way analysis of variance followed by Tukey’s multiple comparison test was used for multiple comparisons. Data are presented as the mean ± standard error (SE). A *p* value of <0.05 was considered statistically significant.

## 3. Results

### 3.1. ATMi and Chk1i Exert Antitumor Effects as Single Agents

To analyze the single-agent effects of the ATMi and Chk1i, HCT116, DLD-1, and HT29 cells were stimulated with each agent alone, and viability was assessed using the WST assay. Each drug reduced the viability of the three colorectal cancer cell lines in a concentration-dependent manner (Figure 1A). The IC_50_ values are shown in Appendix A.

### 3.2. Combined ATMi and Chk1i Treatment Exhibits Synergistic Antitumor Effects

Next, we analyzed the effect of the combined ATMi and Chk1i treatment on cell viability. The concentration of one drug was set to achieve a cell viability of 80% or greater in monotherapy, and the effect of the drug combination was analyzed by setting multiple concentrations of the other drug. In the single-agent group, cell viability decreased only under treatment with high concentrations of the drug, whereas in the drug combination group, cell viability decreased under treatment with lower concentrations of the drug than in the single-agent group (Figure 1B,C). When each drug concentration was compared between the single-drug group and drug combination group, the drug combination group showed significantly lower values. In addition, the combined treatment appeared to reduce cell viability compared with single-agent administration, even when assessed visually (Appendix A). Therefore, we evaluated the combined effect using the CI. The CI values for the three cell types were indicative of a synergistic effect, indicating that ATMi and Chk1i exert synergistic antitumor effects in combination (Figure 1D and Appendix A). The following experiments were performed at concentrations of 10 μM for the ATMi and 0.03 μM for the Chk1i, which showed a specific level of combined efficacy against HCT116 and DLD-1 cells.

SF-TY human normal fibroblasts were used to examine the effect of the ATMi and Chk1i combination on normal cells. Cells were treated with 10 µM ATMi and 0.03 µM Chk1i, the highest concentrations of the drugs used in combination against the three cancer cell types. The viability of SF-TY cells decreased with the combined treatment (Figure 1E and Appendix A).

### 3.3. Combined ATMi and Chk1i Treatment Synergistically Decreases CDK1 Phosphorylation Levels

To characterize the changes in signal transduction induced by the drug combination, we used Western blotting to analyze the activities of Chk1 and Chk2, substrates of ATR and ATM, respectively, and CDK1, which regulates G2/M phase progression. Chk1 phosphorylation at S296 was decreased and increased at S345 in the Chk1i group (Figure 2), indicating that the Chk1i selectively inhibits phosphorylation at S296, the site of action. In addition, phosphorylation at S345 is a marker for ATR activation, and the enhanced phosphorylation at S345 in the Chk1i group indicates that ATR is activated by replication stress and the induction of DNA SSBs. Chk2 phosphorylation was decreased in both the ATMi and combination groups and increased in the Chk1i group (Figure 2), indicating that the Chk1i increases dependence on the ATM pathway and that the ATMi inhibits ATM activity. Furthermore, phosphorylation of CDK1 was significantly decreased in the combination group (Figure 2). CDK1 is a component of the M phase accelerating factor, which is activated by dephosphorylation at T14 and Y15 to promote the transition to mitosis [31]. The levels of T14 and Y15 phosphorylation in CDK1 were lower in the combination group compared with the single-agent group, indicating that the combined treatment activates CDK1, disrupts the G2/M phase checkpoint, and promotes cell cycle progression. In SF-TY cells, unlike cancer cell lines, the level of S296 phosphorylation of Chk1 was barely detectable in Western blotting analyses (Appendix A). However, Chk1i increased the level of S345 phosphorylation of Chk1, and when combined with ATMi, Chk1i slightly decreased the levels of T14 and Y15 phosphorylation of CDK1 in SF-TY (Appendix A), suggesting that modest Chk1 activation also occurs in normal cells. These results also suggest that in normal cells, combination therapy may pass the G2/M checkpoint by activating CDK1.

### 3.4. The ATMi and Chk1i Combination Increases the Number of Cells in the Sub-G1 Phase

We also investigated the effect of the combined treatment with the ATMi and Chk1i on the progression of HCT116 and SF-TY cells through the cell cycle. In addition to determining the percentage change in the number of cells in each phase (G0/G1, S, and G2/M), we also determined the percentage change in the number of cells in the sub-G1 phase, which indicates apoptosis [32]. In HCT116 cells, the number of cells in the G0/G1 phase decreased and the number of cells in the sub-G1 phase significantly increased in the combination group (Figure 3A,B, and Appendix A), suggesting that the drug combination induced cell cycle progression and apoptosis. The number of SF-TY cells in the G0/G1 phase decreased and the number of cells in the S and G2/M phases increased in the combination group, indicating cell cycle arrest at the M phase checkpoint (Figure 3C,D). A total of 10.3% and 11.1% of SF-TY cells were in the sub-G1 phase in the control and combination groups, respectively. The difference was not significant, indicating the drug combination does not induce apoptosis in normal cells.

### 3.5. The ATMi and Chk1i Combination Induces Apoptosis by Introducing DNA Damage

The number of phospho-histone H2A.X (γH2AX; a DNA DSB marker)-positive HCT116 cells was significantly increased in the combination group compared with the other groups (Figure 4A,B). γH2AX intensity was plotted against the DAPI intensity to determine at which phase of the cell cycle DNA damage was induced [33]. A large proportion of γH2AX-positive cells contained a higher DNA content, corresponding to S-phase and especially G2/M phase cells (Appendix A). The number of γH2AX-positive SF-TY cells was increased in the Chk1i and combination groups, as in the HCT116 cells (Appendix A). However, the positivity rate in the combination group was 53.4% in the HCT116 cells, compared with only 6.0% in the SF-TY cells. These results indicate that the ATMi and Chk1i combination induces DSBs in normal cells but at a lower level than in cancer cells. The proportion of cells with multiple 53BP1 foci increased in the ATMi, Chk1i, and combination groups compared with the control group (Figure 4C,D), suggesting that these agents damage DNA. Contrary to expectations, however, the combination therapy resulted in a lower proportion of cells with multiple 53BP1 foci than treatment with Chk1i alone, suggesting that because of the presence of 53BP1 downstream of ATM [34,35], addition of the ATMi in the combination therapy reduced the mobilization of 53BP1 to the site of DNA damage. The number of HCT116 cells with micronuclei was significantly increased following the combined treatment, suggesting that the drug combination induces abnormal DNA division (Appendix A).

TUNEL analyses, which indicate the extent of apoptosis, showed that the drug combination increased the percentage of apoptotic cells (Figure 4E,F). These results suggest that simultaneous inhibition of ATM and Chk1 in colorectal cancer cells induces DNA abnormalities, including DSBs, during cell cycle phases involving a high DNA content, particularly the M phase, leading to cell death through apoptosis.

### 3.6. The ATMi and Chk1i Combination Significantly Reduces Tumor Volume in Syngeneic Tumor Model Mice

We evaluated the effect of the combined ATMi and Chk1i treatment on tumor volume in syngeneic tumor model mice according to previously reported protocols [28,29] (Figure 5A). As it is desirable to administer drugs at the lowest concentration possible to reduce the frequency of side effects, we used concentrations one-tenth of those previously reported (10 mg/kg/day for KU60019 and 3 mg/kg/week for LY2606368) [28,29]. No significant differences in tumor volume between the control group and single-agent groups were observed, but the combination group exhibited a significant decrease in tumor volume compared with the control group (Figure 5B). These results indicate that the ATMi and Chk1i combination is effective against tumors in syngeneic tumor model mice as well as colorectal cancer cell lines.

## 4. Discussion

In the present study, we found that the combined administration of ATMi and Chk1i exhibited synergistic antitumor effects and induced synthetic lethality in colorectal cancer cells. Activation of both ATM and Chk1 commonly arrest the cell cycle by promoting the phosphorylation of CDK1 T14 and Y15. Therefore, we speculated that the phosphorylation level of CDK1 would be decreased by treatment with the combination of inhibitors relative to treatment with the single agents. In the present study, we confirmed that the levels of T14 and Y15 phosphorylation in CDK1 were decreased in the combination group compared with the other groups, indicating that the combined treatment activates CDK1, disrupts the G2/M phase checkpoint, and promotes cell cycle progression. Only HCT116 cells exhibited elevated Chk1 protein levels when treated with Chk1i. This suggests that Chk1 inhibition affects DDR-related proteins differently in different types of colon cancer cells. The number of HCT116 cells in the G0/G1 phase decreased and the number of cells in the sub-G1 phase increased significantly in the combination group, suggesting that the drug combination induced apoptosis by causing cells to skip the cell cycle checkpoints. The combination group had more γH2AX-positive cells than the other groups, but the number of multiple 53BP1 foci-positive cells was decreased compared with the Chk1 group. It has been reported that ATM regulates 53BP1 activity [35] and that 53BP1 foci are decreased by ATM inhibitors [34]. Therefore, in combination therapy, the accumulation of 53BP1 into sites of DNA damage by the Chk1i can be suppressed by the ATMi. Thus, the inhibitory effect of ATMis on the activity of 53BP1 involved in DNA repair processes, such as homologous recombination, may prevent the repair of DNA damage caused by the Chk1i and help efficiently induce cell death. Although the detailed mechanism of cell death could not be elucidated in this study, the number of TUNEL-positive cells and sub-G1 cells increased in the combination group, suggesting that DNA damage is followed by apoptosis induction, leading to cell death.

Inhibition of Chk1 resulted in the accumulation of DNA damage, suggesting that Chk1 plays an important role not only in regulating the cell cycle checkpoint but also in suppressing replication origin firing in response to replication stress [26]. Cancer cells are reportedly subjected to increased replication stress through various mechanisms and rely on the Chk1 pathway for survival, which suppresses replication stress [36]. In the present study, colon cancer cell lines also clearly exhibited higher levels of S296 phosphorylation, a marker of Chk1 activation, compared with normal SF-TY cells in the absence of any treatment. This finding is consistent with the finding that cancer cells activate Chk1 to suppress replication stress and DNA damage [36]. Because inhibition of Chk1 is known to promote excessive replication origin firing, replication fork stalling, and the formation of single-stranded DNA [37,38], Chk1 inhibition can increase replication stress in cancer cells, resulting in DSBs. In the combined treatment, inhibition of ATM may have reduced the cellular response to DSBs and thereby induced DSBs in a larger number of cancer cells. A previous report demonstrated that simultaneous inhibition of Chk1 and ATM increases cell death in NB-39-nu and SK-N-BE neuroblastoma cells, which are not generally responsive to Chk1 inhibitors [39]. The report concluded that suppression of the ATM-p53-p21 axis, which is activated by Chk1 inhibition, via the inhibition of ATM leads to a synergistic antitumor effect. Another study reported that inhibition of ATR is effective against cancer cells in ATM-defective chronic lymphocytic leukemia [27]. These findings in conjunction with those of our present study indicate that simultaneous inhibition of the SSB and DSB pathways in cancer cells is lethal to cancer cells under high replication stress. Currently, in addition to Chk1 and ATM, some DDR-related components have been extensively studied, such as PARP, ATR, ATM, Chk1, Wee1, and DNA-dependent protein kinase catalytic subunit (DNA-PKcs) [40]. It has been shown that inhibition of multiple factors involved in the DDR dramatically increases its effectiveness, which is consistent with the present results.

As Chk1 and ATM are located upstream of the two pathways that constitute the cell cycle checkpoint, inhibition of Chk1 and ATM could adversely affect normal cells. In the WST assay, the viability of SF-TY cells decreased similar to cancer cells, but the percentage of cells in the sub-G1 phase in the cell cycle analysis was similar to that in the control group, indicating that the decrease in cell viability was not due to cell death via apoptosis. In the DNA damage analysis, the percentage of γH2AX-positive cells in the Chk1i and combination groups increased, suggesting that inhibition of Chk1 induces significant replication stress, leading to an increase in the frequency of damage even in normal cells. However, the proportion of γH2AX-positive SF-TY cells was one-tenth lower than that of HCT116 cells in the combination group, suggesting that DNA damage is less likely to occur in normal cells than cells undergoing oncogene-induced replication stress. The combination therapy tended to reduce the levels of phosphorylated CDK1 in SF-TY and colon cancer cell lines, although not significantly. Therefore, the combination therapy can facilitate passage of the G2/M phase checkpoint in normal cells, such as SF-TY cells. The cell cycle analysis revealed an increase in the percentage of cells in the S and G2/M phases, suggesting that the cell cycle is arrested at the M phase checkpoint after passing the G2/M checkpoint in normal cells. The M phase checkpoint is regulated by molecules different from ATM and Chk1 [41], and it is thought that normal cells in which these factors are functioning properly can readily detect minute DNA damage and arrest the cell cycle.

Among DDR inhibitors, Chk1 inhibitors for cancer therapy have been well studied. LY2606368 (prexasertib) is an ATP-competitive inhibitor with high selectivity for Chk1. The Chk1i LY2606368 is currently undergoing clinical trials. In a phase I study of LY2606368 as a single agent in patients with advanced solid tumors, serious adverse effects involving myeloid cells were observed, such as neutropenia, thrombocytopenia, and anemia [42]. On the other hand, studies using cultured cells and animals have shown the efficacy of LY2606368 [33,43]. The results of ongoing clinical trials will clarify the efficacy of Chk1i in various cases. KU-60019, an analogue of the first selective ATM inhibitor KU-55933, is a competitive ATM inhibitor with improved pharmacokinetics and bioavailability compared with KU-55933 [40]. In glioblastoma cells, KU-60019 interrupts DDR signals by efficiently blocking ATM activity and downregulates AKT phosphorylation to reduce cell survival [44]. In phosphatase and tensin homolog deleted from chromosome 10 (PTEN)-deficient breast cancer cells, KU-60019 combined with cisplatin induced synthetic lethality with increasing γH2AX foci and PARP cleavage [45]. In addition, KU-60019 was shown to selectively damage PTEN-deficient colorectal cancer cells both in vitro and in vivo [29]. Thus, KU-60019 has shown promising results in preclinical studies. KU-60019 is currently undergoing clinical trials using organotypic cultures derived from kidney cancer patients (NCT035714389) [40]. An orally available ATMi, AZD0156, is also currently undergoing clinical trials. While the combination with Oriparab or other anticancer drugs can be expected to demonstrate anticancer effects, it was suggested that high doses can cause hematologic adverse effects, including neutropenia, thrombocytopenia, and anemia, similar to Chk1is [46]. The results of this study suggest that combination therapy involving a Chk1i and ATMi would exhibit fewer side effects while providing sufficient anticancer activity by reducing the dosage of the individual drugs.

The HCT116, DLD-1, and CT26 cells in which synergistic effects were shown in this study are colon cancer cells harboring K-ras mutations [47,48]. Approximately 50% of colorectal cancer patients have RAS mutations, but anti-EGFR antibodies such as cetuximab and panitumumab, which are used in the treatment of colorectal cancer, are effective only in patients without RAS mutations, thus highlighting the need for colorectal cancer treatments for patients with RAS mutations. Our findings suggest that the combined ATMi and Chk1i treatment could be effective in colorectal cancer patients with K-ras mutations. Chk1 inhibition in p53-deficient cell lines induces chromosome fragmentation and caspase-independent cell death [49]. Moreover, ATM inhibition with ionizing radiation significantly prolongs p53 mutant glioblastoma mouse survival [50]. In our study, the Chk1i was particularly potent against HT29 cells with the p53 mutation, and the ATMi also seemed to be slightly more effective against HT29 cells at a certain concentration. The drug combination was shown to be more effective against HT29 cells than either single agent as well as more effective against the other two cell types. The concentrations that showed efficacy against HT29 cells in the drug combination group were lower than those effective against the other two cell types. These results suggest that both the single-agent and combination treatments are more effective against cancers with p53 mutations than against cancers with wild-type p53. However, it was also reported that low concentrations of KU-59403 induce sensitivity to chemotherapeutic agents or IR in human cancer cells, regardless of the presence or absence of p53 mutations [51]. The study concluded that p53 mutations are not correlated with ATM inhibition sensitivity. Therefore, further studies will be needed regarding the sensitivity to ATM inhibition and the presence of p53 mutations. It is quite possible that the activation state of other factors involved in DDR besides p53 and the degree of replication stress may be responsible for drug sensitivity to combined therapy. We would like to clarify in future studies which mutations in colorectal cancer cells are more or less likely to interfere with the efficacy of the combined treatment.

There were some limitations to this study. First, the antitumor effect in the in vivo model of the combined treatment was limited. Given that in vitro inhibition of Chk1 and ATM produced significant antitumor effects, it will be necessary to determine the optimal dose and route of administration that will produce the combined effect or use a different inhibitor that has superior pharmacokinetics. Preclinical studies using mice will be necessary in the future to determine whether the combination therapy has potential as a new anticancer therapy. Second, we were unable to elucidate whether long-term drug combination treatment has any kind of effect on normal tissues, such as aging or secondary cancers. Therefore, more detailed analyses using in vivo models will be needed to advance the combined treatment to the clinical application. Third, we were unable to clarify whether the combined treatment is effective only for colorectal cancer or for other cancers as well. Given the mechanism of the combination therapy, it may exert antitumor effects for other carcinomas exhibiting genomic instability. We plan to address these issues in future studies.

## 5. Conclusions

We demonstrated that the combined ATMi and Chk1i treatment exerts synergistic antitumor effects and induces synthetic lethality in colorectal cancer cells at low doses. The ATMi and Chk1i combination synergistically promotes CDK1 activation by suppressing its phosphorylation at T14 and Y15. Activation of cell cycle regulators, including CDK1, induces cell death through apoptosis by enhancing cell cycle progression in cancer cells harboring DNA abnormalities, including DSBs generated by the inhibition of SSB and DSB repair processes (Figure 6). Considering that the drug combination exerts antitumor effects at one-tenth the concentrations described in previous reports, it could be a valuable treatment exhibiting high antitumor efficacy and few side effects. We hope that these findings will lead to the development of new treatments for colorectal cancer that exploit the inherent genomic instability of cancer cells.

## Figures and Tables

**Figure 1 cancers-15-00735-f001:**
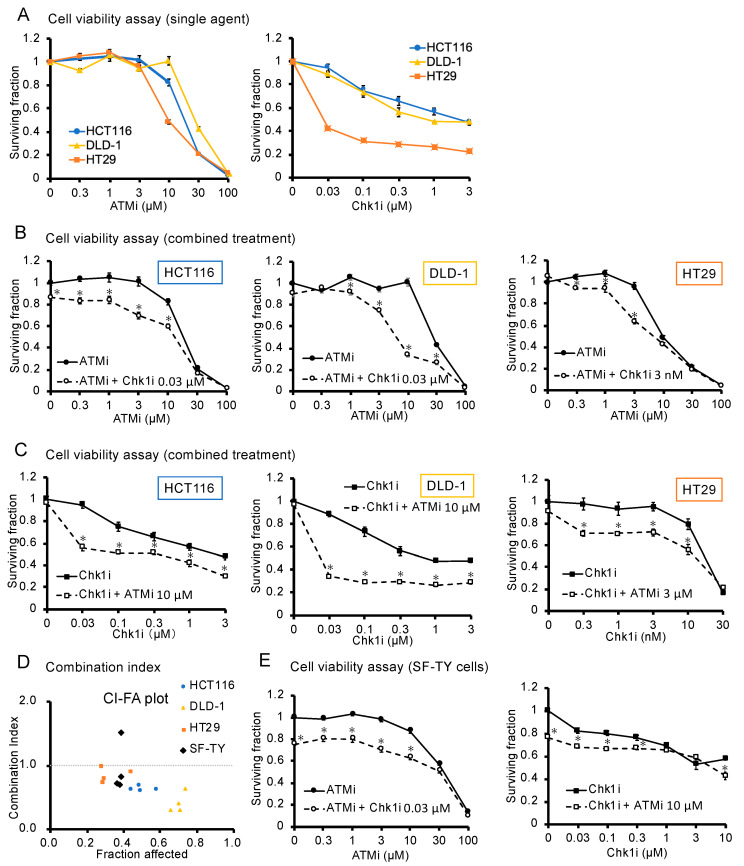
Analysis of the antitumor effect of the ATMi and Chk1i combination. (**A**) Evaluation of the cytotoxicity of the ATMi and Chk1i after 48 h of treatment, as determined using the WST assay. The viability of the untreated cells was defined as 1. Data are presented as mean ± SE (*n* = 5). (**B**,**C**) Effect of the combined ATMi and Chk1i treatment on HCT116, DLD-1, HT29, and SF-TY cells evaluated after 48 h using the WST assay. The viability of the untreated cells was defined as 1. Data are presented as mean ± SE (*n* = 5; * *p* < 0.05; Tukey’s multiple comparison test). (**D**) Combination index (CI) values for HCT116, DLD-1, HT29, and SF-TY cells were calculated based on the WST assay results and are shown in the graph with fraction affected (FA), where a CI < 1 indicates a synergistic effect, CI = 1 indicates an additive effect, and CI > 1 indicates an antagonistic effect. (**E**) Effect of the combined ATMi and Chk1i treatment on SF-TY cells evaluated after 48 h using the WST assay. The viability of the untreated cells was defined as 1. Data are presented as mean ± SE (*n* = 5; * *p* < 0.05 compared to single drug stimulation group).

**Figure 2 cancers-15-00735-f002:**
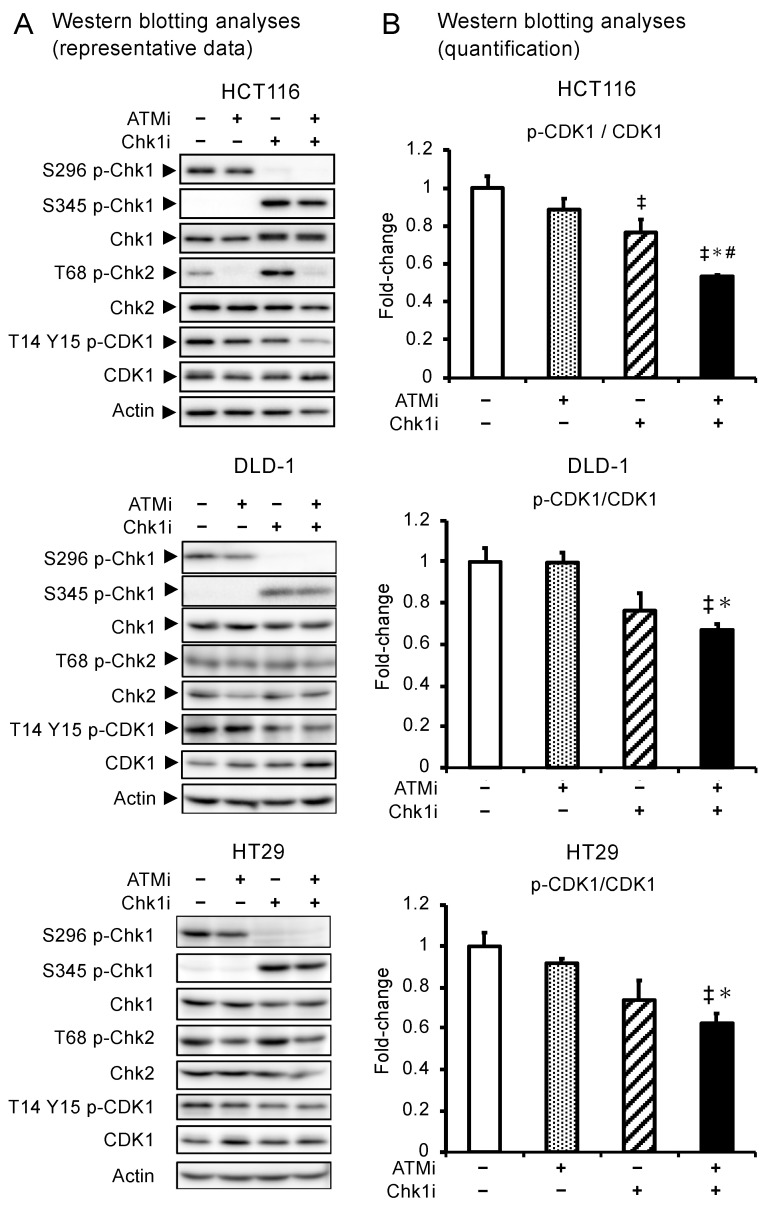
Phosphorylation of Chk1, Chk2, and CDK1 during drug treatment. (**A**) Western blotting analysis of Chk1, phospho-Chk1 (S296 and S345), Chk2, phospho-Chk2 (T68), CDK1, and phospho-CDK1 (T14 and T15) in HCT116, DLD-1, and HT29 cells treated with ATMi (10 μM), Chk1i (0.03 μM), or both drugs for 6 h. (**B**) Quantification of the amount of each protein as determined by Western blotting analysis in the three cancer cell lines. Protein levels were normalized to actin levels. Data are presented as mean ± SE (*n* = 3; ^‡^ *p* < 0.05, vs. control; * *p* < 0.05, vs. ATMi group, # *p* < 0.05, vs. Chk1i group; Tukey’s multiple comparison test). Uncropped versions of the immunoblotting data are shown in Appendix A. Relative intensities of the bands in the immunoblot data are shown in Appendix A.

**Figure 3 cancers-15-00735-f003:**
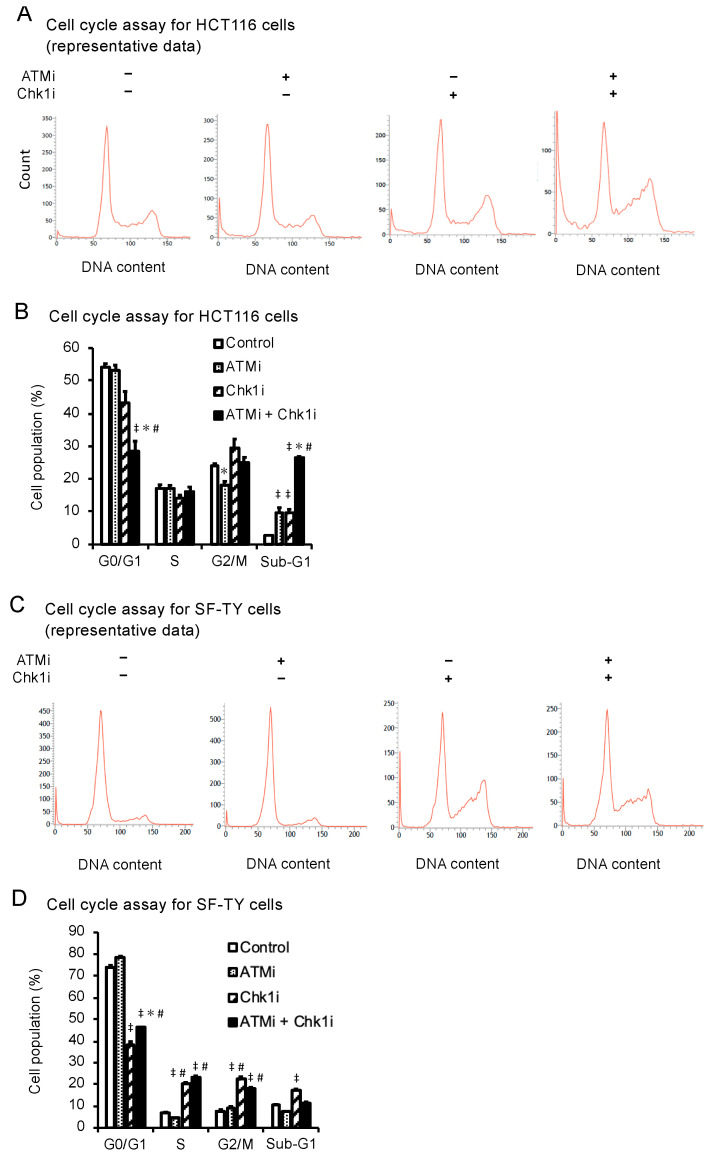
Analysis of the cell cycle progression of cancer and normal cells under the combined ATMi and Chk1i treatment. (**A**) Representative results of the cell cycle progression of HCT116 cells treated with ATMi (10 μM), Chk1i (0.03 μM), or both drugs for 48 h, as determined by flow cytometry using PI staining. (**B**) Percentage of HCT116 cells in each cell cycle phase, as determined by PI staining. Data are presented as mean ± SE (*n* = 3; ^‡^
*p* < 0.05, vs. control; * *p* < 0.05, vs. ATMi group, # *p* < 0.05, vs. Chk1i group; Tukey’s multiple comparison test). (**C**) Representative results of the cell cycle progression of SF-TY cells treated with ATMi (10 μM), Chk1i (0.03 μM), or both drugs for 48 h, as determined by flow cytometry using PI staining. (**D**) Percentage of SF-TY cells in each cell cycle phase, as determined by PI staining. Data are presented as mean ± SE (*n* = 3; ^‡^
*p* < 0.05, vs. control; * *p* < 0.05, vs. ATMi group, # *p* < 0.05, vs. Chk1i group; Tukey’s multiple comparison test).

**Figure 4 cancers-15-00735-f004:**
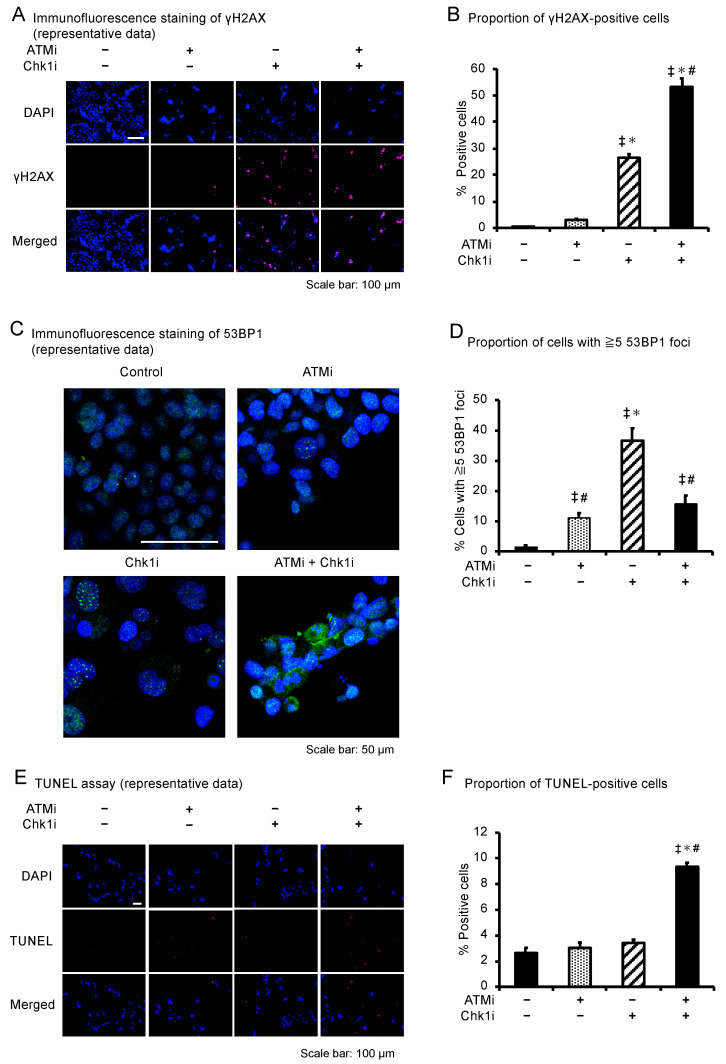
DNA damage and apoptosis of cancer cells under the combined ATMi and Chk1i treatment. (**A**) Immunofluorescence staining of γH2AX (pink) in HCT116 cells treated with ATMi (10 μM), Chk1i (0.03 μM), or both drugs for 48 h. DAPI = blue. Scale bars = 100 μm. (**B**) Ratio (%) of γH2AX-positive HCT116 cells. Data are presented as mean ± SE (*n* = 6; 6 fields/well; ^‡^
*p* < 0.05, vs. control; * *p* < 0.05, vs. ATMi group, ^#^
*p* < 0.05, vs. Chk1i group). (**C**) Immunofluorescence staining of 53BP1 (green) in HCT116 cells treated with ATMi (10 μM), Chk1i (0.03 μM), or both drugs for 48 h. DAPI = blue. Scale bars = 50 μm. (**D**) Ratio (%) of HCT116 cells with ≥5 53BP1 foci. Data are presented as mean ± SE (*n* = 3; 30 cells/well; ^‡^
*p* < 0.05, vs. control; * *p* < 0.05, vs. ATMi group, ^#^
*p* < 0.05, vs. Chk1i group). (**E**) TdT-mediated dUTP nick-end labeling (TUNEL) analysis of HCT116 cells treated with ATMi (10 μM), Chk1i (0.03 μM), or both drugs for 24 h. TUNEL = red; DAPI = blue. Scale bars = 50 μm. (**F**) Ratio (%) of TUNEL-positive HCT116 cells. Data are presented as mean ± SE (*n* = 6; 6 fields/well; ^‡^
*p* < 0.05, vs. control; * *p* < 0.05, vs. ATMi group, ^#^
*p* < 0.05, vs. Chk1i group).

**Figure 5 cancers-15-00735-f005:**
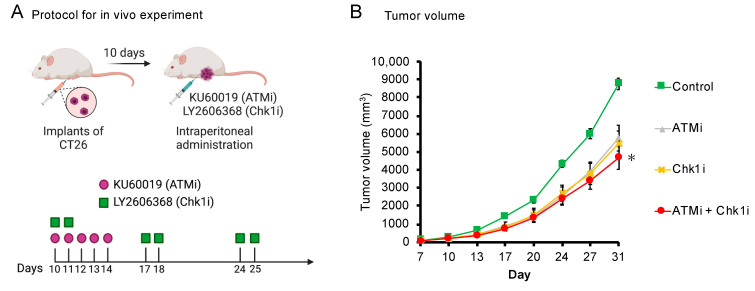
Effect of the combined ATMi and Chk1i treatment on syngeneic tumor model mice. (**A**) Protocol for the analysis of the effect of the combined ATMi and Chk1i treatment on syngeneic tumor model mice. CT26 cells were implanted subcutaneously. (**B**) Tumor volume of the control, ATMi, Chk1i, and ATMi + Chk1i groups was monitored for 31 days. Data are presented as mean ± SE (control group: *n* = 5, ATMi group: *n* = 7, Chk1i group: *n* = 3, ATMi + Chk1i group: *n* = 4; * *p* < 0.05, vs. control on day 31; Tukey’s multiple comparison test).

**Figure 6 cancers-15-00735-f006:**
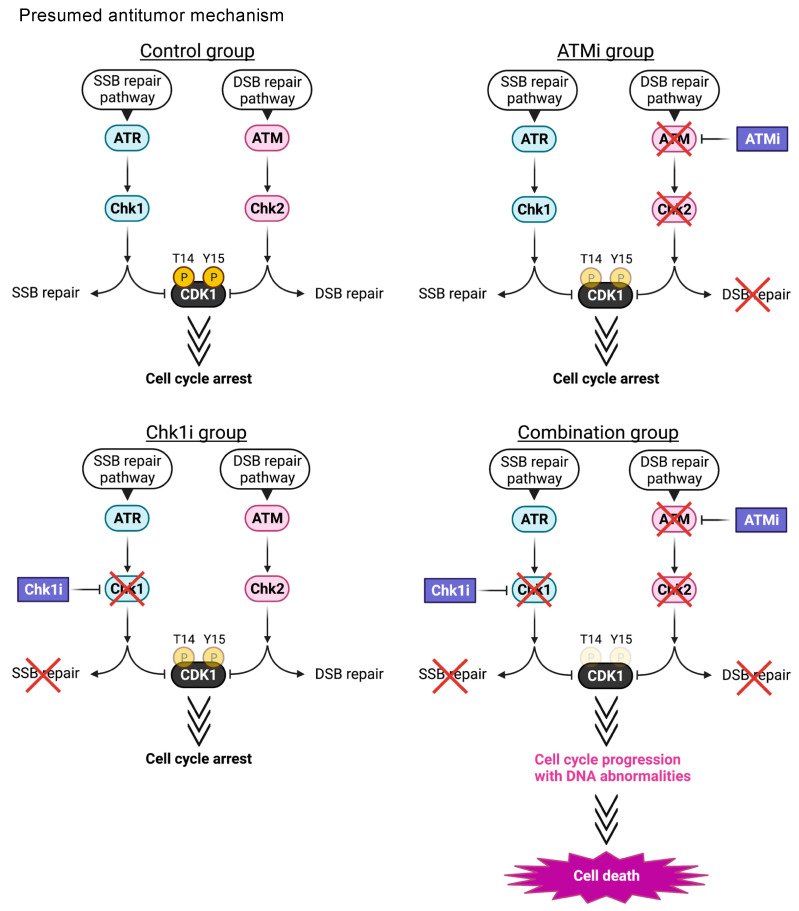
Presumed antitumor mechanism. Putative mechanism of ATMi- and Chk1i-induced synthetic lethality in colorectal cancer cells.

## Data Availability

The data presented in this study are available in this article (and Appendix A).

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
