# Peer review of "The Combination of ATM and Chk1 Inhibitors Induces Synthetic Lethality in Colorectal Cancer Cells"

_cancers, 2023, doi:10.3390/cancers15030735_

Round 1
Reviewer 1 Report (Previous Reviewer 2)
The new revised manuscript addresses the earlier concerns raised and has greatly improved the quality of the manuscript.
This manuscript is a resubmission of an earlier submission. The following is a list of the peer review reports and author responses from that submission.
Round 1
Reviewer 1 Report
The authors investigated the association of ATM and ChK1 inhibitors in colorectal cancer cells. The theoretical basis for this association was synthetic lethality, which is an important approach to controlling cancer cells. The work is important, but the text has some deficiencies.
Please, find hereby, a point-by-point comment list to the manuscript.
· Abstract. Lines 40 – 43. The sentence is too long and it is not necessary to explain that phosphorylated H2AX is a marker of DNA damage or that TUNEL-marked cells are apoptotic. It's redundant. Please rewrite objectively.
· Introduction. Page 2, 2nd paragraph, line 75. The sentence “CDC25 activates CDK1 by dephosphorylating T14 and Y15” can be rewritten as “CDC25 activates CDK1 by dephosphorylation of T14 and Y15”.
· Introduction. Page 2, 4th paragraph, lines 97-98. The definition of synthetic lethality should be improved. It is not just based in gene deletions. The synthetic lethality involves interaction between specific genes or common molecular pathways deregulated during carcinogenesis. Please see Wu X, Lippman SM. Nature Reviews Cancer (2011) 11, 879-885.
· Materials and Methods. Page 3, line 132. Why the cells were incubated only at 48 hours? Were no treatments performed at other time intervals (24, 72h)?
· Materials and Methods. Page 3, line 133. What is CCK-8? Is it a WST-8 assay reagent? This has to be better described.
· Materials and Methods. Page 3, line 145. Western blotting analyses instead of analysis.
· Materials and Methods. Page 4, line 161. “Analyses” instead of analysis.
· Materials and Methods. Page 4, line 162. NaN3 instead of NaN3.
· Materials and Methods. Page 4, line 203. “In vivo experiment” instead BALB/c mice.
· Materials and Methods. Page 5, line 207. Why did the authors use female mice?
· Materials and Methods. Page 5, line 208. What is CT26? There is no previous mention in the text. If they are cells, then they should have been described along with the other cells used in the work. This description must contain the origin and conditions of culture.
· Materials and Methods. Page 5, line 208.“…received subcutaneous implants of 1.0x104 CT26 cells/µL” instead of were subcutaneously treated with 2.0x106 cells/200 µL of CT26 cells.
· Results. Page 5, line 239. “(Figures 1B and 1C) instead of (Figure 1B,C).
· Figure 1 needs to be formatted. Each graph must have a short title. The ordinate axis of all graphs must have a legend and not just on the first graph of each item. Other figures: all of them must have a short title.
· Figure 4. CT26 cells implants instead injection of CT26. Were the implants subcutaneous? This must be described in the legend
· Results. Page 9, line 318-319. “(Figures 3A, 3B and Supplementary Figure S2) instead of (Figure 3A, B, Supplementary Figure S2).
· Results. Page 5, line 343. “(Figures 4C and 4D) instead of (Figure 4C,D).
· Discussion. 1st Paragraph. Lines-393-411. In this paragraph, the authors should establish the link between the hypotheses established in the introduction with the main findings of the work. However, the authors presented only a summary of the results already presented. This paragraph must be rewritten.
· Discussion. 3rd Paragraph. Please remove (Figure 3C,D), (Supplementary Figure S4), (Figure 4A,B, Supplementary Figure S4), and (Figure 3C, D).
· Conclusion. Line 516 “We” (capital letter).
· Supplementary Materials. Components. Three Supplementary Tables (plural)
Author Response
Manuscript ID: cancers-1873950
Tozaki et al. “The combination of ATM and Chk1 inhibitors induces synthetic lethality in colorectal cancer cells”
REVIEWER 1
We wish to express our sincere gratitude to the reviewer for these insightful comments, which helped us significantly improve the manuscript.
- Abstract. Lines 40 – 43. The sentence is too long and it is not necessary to explain that phosphorylated H2AX is a marker of DNA damage or that TUNEL-marked cells are apoptotic. It's redundant. Please rewrite objectively.
We have deleted these sentences in the Abstract (p. 1).
- Introduction. Page 2, 2nd paragraph, line 75. The sentence “CDC25 activates CDK1 by dephosphorylating T14 and Y15” can be rewritten as “CDC25 activates CDK1 by dephosphorylation of T14 and Y15”.
We have revised the text in the Introduction as follows (p. 2): " CDC25 activates CDK1 by dephosphorylating T14 and Y15" to: " CDC25 activates CDK1 by dephosphorylation of T14 and Y15".
- Introduction. Page 2, 4th paragraph, lines 97-98. The definition of synthetic lethality should be improved. It is not just based in gene deletions. The synthetic lethality involves interaction between specific genes or common molecular pathways deregulated during carcinogenesis. Please see Wu X, Lippman SM. Nature Reviews Cancer (2011) 11, 879-885.
We have revised the text as follows (p. 2-3): "The phenomenon of lethality associated with multiple concurrent gene deletions is called "synthetic lethality". For example, synthetic lethality can be induced by poly (ADP-ribose) polymerase 1 (PARP1) inhibitors in breast and ovarian cancer cells with inactivating mutations in the BRCA1/2 genes [22]." to ""Synthetic lethality" is defined as a type of genetic interaction in which the combination of more than two genetic events results in cell death. Synthetic lethality is thought to kill cancer cells specifically without affecting normal cells by acting on specific genes or common molecular pathways regulated in the carcinogenesis process. For example, synthetic lethality can be induced by poly (ADP-ribose) polymerase 1 inhibitors in breast and ovarian cancer cells with inactivating mutations in the BRCA1/2 genes [22]. Another example of selective killing of cancer cells is the treatment of non-small cell lung cancer cells with driver mutations in Kirsten rat sarcoma viral oncogene homologue (K-ras) with tumor necrosis factor-related apoptosis-inducing ligand and mimic of second mitochondria-derived activator of caspase [23]. " We have also added the following reference (p. 17): “23. Wu, X.; Lippman, S.M. An intermittent approach for cancer chemoprevention. Nat. Rev. Cancer. 2011, 11, 879-885.” Your suggestions helped us greatly improve our manuscript.
- Materials and Methods. Page 3, line 132. Why the cells were incubated only at 48 hours? Were no treatments performed at other time intervals (24, 72h)?
As there were many combinations of cell types and drug concentrations examined in the cell toxicity assay, we conducted experiments for 48 h, a time period that has generally been extensively evaluated. Although data are not shown in this manuscript, it was visually confirmed that there was minimal impact on cell survival at 24 h.
- Materials and Methods. Page 3, line 133. What is CCK-8? Is it a WST-8 assay reagent? This has to be better described.
CCK-8 stands for “cell counting kit-8”, but this is not a common name. Therefore, we have revised the text as follows (p. 3): "CCK-8 solution" was changed to “WST-8 assay reagent".
- Materials and Methods. Page 3, line 145. Western blotting analyses instead of analysis.
We have revised the text as follows (p. 3): "analysis" was changed to "analyses".
- Materials and Methods. Page 4, line 161. “Analyses” instead of analysis.
We have revised the text as follows (p. 4): "analysis" was changed to "analyses".
- Materials and Methods. Page 4, line 162. NaN3 instead of NaN3.
We have revised the text as follows (p. 4): "NaN3" was changed to "NaN3".
- Materials and Methods. Page 4, line 203. “In vivo experiment” instead BALB/c mice.
We have revised the text as follows (p. 5): "BALB/c mice" was changed to "In vivo experiment".
- Materials and Methods. Page 5, line 207. Why did the authors use female mice?
We thank the Reviewer for this question. Indeed, it may have been unnatural to use only female mice in the experiment. We used only female mice in this study because female mice are often used for mouse tumor models.
- Materials and Methods. Page 5, line 208. What is CT26? There is no previous mention in the text. If they are cells, then they should have been described along with the other cells used in the work. This description must contain the origin and conditions of culture.
We thank the Reviewer for pointing out the omission, which we did not notice. CT26 cells were purchased from the same source and cultured in the same way as HCT116 and HT29 cells. We have added information about CT26 cells in the Materials and Methods section (p. 3).
- Materials and Methods. Page 5, line 208.“…received subcutaneous implants of 1.0x104 CT26 cells/µL” instead of were subcutaneously treated with 2.0x106 cells/200 µL of CT26 cells.
We have revised the text as follows (p. 5): "were subcutaneously treated with 2.0x106 cells/200 µL of CT26 cells" was changed to "received subcutaneous implants of 1.0x104 CT26 cells/µL".
- Results. Page 5, line 239. “(Figures 1B and 1C) instead of (Figure 1B,C).
We have revised the text as suggested.
- Figure 1 needs to be formatted. Each graph must have a short title. The ordinate axis of all graphs must have a legend and not just on the first graph of each item. Other figures: all of them must have a short title.
We have added a short title and ordinate axis to each figure.
- Figure 4. CT26 cells implants instead injection of CT26. Were the implants subcutaneous? This must be described in the legend
CT26 cells were implanted subcutaneously. We have added the following text in the Legends section (p. 11): “CT 26 cells were implanted subcutaneously."
- Results. Page 9, line 318-319. “(Figures 3A, 3B and Supplementary Figure S2) instead of (Figure 3A, B, Supplementary Figure S2).
We have revised the text as suggested.
- Results. Page 5, line 343. “(Figures 4C and 4D) instead of (Figure 4C,D).
We have revised the text as suggested.
- Discussion. 1st Paragraph. Lines-393-411. In this paragraph, the authors should establish the link between the hypotheses established in the introduction with the main findings of the work. However, the authors presented only a summary of the results already presented. This paragraph must be rewritten.
We agree completely with your comment and have revised the first paragraph as suggested (p. 11-12): " In this study, we found that combined administration of ATMi and Chk1i exhibited synergistic antitumor effects and induced synthetic lethality in colorectal cancer cells. The drug combination exerted antitumor effects at very low concentrations compared with single-agent treatment (Figure 1B, C). The synergistic effect of the ATMi and Chk1i combination was confirmed by CI values (Figure 1D, Supplementary Table S3). The levels of T14 and Y15 phosphorylation in CDK1 decreased in the combination group compared with the other groups, indicating that combined treatment activates CDK1, disrupts the G2/M phase checkpoint, and promotes cell cycle progression (Figure 2). The number of HCT116 cells in the G0/G1 phase decreased and the number of cells in the sub-G1 phase increased significantly in the combination group, suggesting that the drug combination induced apoptosis by causing cells to skip the G1/S and G2/M phase checkpoints (Figure 3A, B). The number of γH2AX-positive HCT116 cells was significantly higher in the combination group versus the other groups (Figure 4A, B). TUNEL staining showed that the combined treatment increased the apoptosis of cancer cells (Figure 4C, D). Based on these results, we propose the following mechanism for the antitumor effect of the combined treatment. The ATMi and Chk1i synergistically promote cell cycle progression via various pathways, including activation of CDK1 via suppression of its phosphorylation. Cancer cells that rotate through the cell cycle without engaging SSB and DSB repair processes accumulate DNA abnormalities, including DSBs, resulting in cell death through apoptosis. " to " In the present study, we found that combined administration of ATMi and Chk1i exhibited synergistic antitumor effects and induced synthetic lethality in colorectal cancer cells. Activation of both ATM and Chk1 commonly arrest the cell cycle by promoting phosphorylation of CDK1 T14 and Y15. Therefore, we speculated that the phosphorylation level of CDK1 would be decreased by treatment with the combination of inhibitors relative to treatment with the single agents. In the present study, we confirmed that the levels of T14 and Y15 phosphorylation in CDK1 were decreased in the combination group compared with the other groups, indicating that combined treatment activates CDK1, disrupts the G2/M phase checkpoint, and promotes cell cycle progression. Only HCT116 cells exhibited elevated Chk1 protein levels when treated with Chk1i. This suggests that Chk1 inhibition affects DDR-related proteins differently in different types of colon cancer cells. The number of HCT116 cells in the G0/G1 phase decreased and the number of cells in the sub-G1 phase increased significantly in the combination group, suggesting that the drug combination induced apoptosis by causing cells to skip the cell cycle checkpoints. The combination group had more γH2AX-positive cells than the other groups, but the number of multiple 53BP1 foci–positive cells was decreased compared with the Chk1 group. It has been reported that ATM regulates 53BP1 activity [35] and that 53BP1 foci are decreased by ATM inhibitors [34]. Therefore, in combination therapy, the accumulation of 53BP1 into sites of DNA damage by the Chk1i can be suppressed by the ATMi. Thus, the inhibitory effect of ATMis on the activity of 53BP1 involved in DNA repair processes, such as homologous recombination, may prevent the repair of DNA damage caused by the Chk1i and help efficiently induce cell death. Although the detailed mechanism of cell death could not be elucidated in this study, numbers of TUNEL-positive cells and sub-G1 cells were increased in the combination group, suggesting that at least DNA damage is followed by apoptosis induction, leading to cell death." Your suggestions have helped us greatly improve our manuscript.
- Discussion. 3rd Paragraph. Please remove (Figure 3C,D), (Supplementary Figure S4), (Figure 4A,B, Supplementary Figure S4), and (Figure 3C, D).
We have removed the indicated text.
- Conclusion. Line 516 “We” (capital letter).
We have revised the text as suggested.
- Supplementary Materials. Components. Three Supplementary Tables (plural)
We have revised the text as suggested.
Thank you again for your valuable comments regarding our manuscript. We hope that the revised manuscript is now suitable for publication.

Reviewer 2 Report
While authors have provided evidences to suggest that the combination of ATM and Chk1 inhibitors is much more potent than single agents, the study does lack a few important controls. For example,
1.The authors should also demonstrate the impact of single and combination drug treatment on the control SF-TY cells in figure 2 (Western blots).
2.Why do the authors see an increase in overall Chk1 levels when treated with Chk1 inhibitor in Figure 2A?
3.What is the possible explanation for Chk1 S296 induction even in untreated HCT116 cells?
4. The authors should have also included pH2Ax, cleaved caspases in their western blot to demonstrate activation of DNA damage and apoptosis.
5.The authors have not mentioned the concentration of ATMi and Chk1i used for H2Ax IF assay.
6.Do the authors also see an increase in 53BP1 foci as an indicator of unresolved DNA damage.
7. The data to evaluate impact of combination treatment on tumor volume seems over exaggerated especially given that the data fig 5B does not suggest a significant difference single agent treatment and combination treatment. So to conclude that the combination treatment demonstrates significant potency may not be the correct interpretation.
Author Response
Manuscript ID: cancers-1873950
Tozaki et al. “The combination of ATM and Chk1 inhibitors induces synthetic lethality in colorectal cancer cells”
REVIEWER 2
We wish to express our sincere gratitude to the reviewer for these insightful comments, which helped us significantly improve the manuscript.
- The authors should also demonstrate the impact of single and combination drug treatment on the control SF-TY cells in figure 2 (Western blots).
We conducted Western blotting using SF-TY cells, and the results are shown in Figure S2. In addition, we have added the following text in the Results section (p. 7): " In SF-TY cells, unlike cancer cell lines, the level of S296 phosphorylation of Chk1 was barely detectable in Western blotting analyses (Supplementary Figure S2A). However, Chk1i increased the level of S345 phosphorylation of Chk1, and when combined with ATMi, Chk1i slightly decreased the levels of T14 and Y15 phosphorylation of CDK1 in SF-TY (Supplementary Figure S2A and S2B), suggesting that modest Chk1 activation also occurs in normal cells. These results also suggest that in normal cells, combination therapy may pass the G2/M checkpoint by activating CDK1." Your suggestion has helped us greatly improve our manuscript.
- Why do the authors see an increase in overall Chk1 levels when treated with Chk1 inhibitor in Figure 2A?
We thank the Reviewer for this question. Your comment made us aware of this issue. We reviewed several papers and found a few that appeared to show an increase in Chk1 protein levels with Chk1 inhibitors, but basically no change. Chk1 inhibitors may increase Chk1 protein levels in HCT116 cells through a feedback effect. We added the following text in the Discussion section (p. 12): "Only HCT116 cells exhibited elevated Chk1 protein levels when treated with Chk1i. This suggests that Chk1 inhibition affects DDR-related proteins differently in different types of colon cancer cells."
- What is the possible explanation for Chk1 S296 induction even in untreated HCT116 cells?
We thank the Reviewer for this question. From reviewing the data in other papers, it appears that CHK1 S296 in cancer cells is phosphorylated to some extent even in the absence of any treatment. Given that phosphorylation of CHK1 S296 was at undetectable levels in normal cells, we speculate that DNA aberrations occur continually in cancer cells and that Chk1 may be activated. We have added the following text in the Discussion section (p. 12): " In the present study, colon cancer cell lines also exhibited clearly higher levels of phosphorylation of S296, a marker of Chk1 activation, compared with normal SF-TY cells in the absence of any treatment. This finding is consistent with the finding that cancer cells activate Chk1 to suppress replication stress and DNA damage [36]. " .
- The authors should have also included pH2Ax, cleaved caspases in their western blot to demonstrate activation of DNA damage and apoptosis.
We agree with your comment. We confirmed by immunofluorescence staining that pH2Ax is increased by the combination therapy, but we could not confirm that cleaved caspase 3 is increased by Western blotting. Some apoptotic events, such as mitotic catastrophe and TNFα-mediated cell death, may not be detected by monitoring early apoptotic markers. In the present study, the combination therapy increased pH2Ax, sub-G1 cells, and TUNEL-positive cells, suggesting that it induces apoptosis. However, it is not clear whether this apoptosis was caspase 3–independent due to DNA damage.
- The authors have not mentioned the concentration of ATMi and Chk1i used for H2Ax IF assay.
We have added the drug concentrations in legends for Figures 2, 3, 4, S2, S4, S6, and S7.
- Do the authors also see an increase in 53BP1 foci as an indicator of unresolved DNA damage.
We conducted immunofluorescence analysis of 53BP1 and included the results in Figure 4. In addition, we have added the following text in the Results section (p. 8 to 10): " The proportion of cells with multiple 53BP1 foci was increased in the ATMi group, Chk1i group, and combination group compared with the control group (Figure 4C and 4D), suggesting that these agents damage DNA. Contrary to expectations, however, the combination therapy resulted in a lower proportion of cells with multiple 53BP1 foci than treatment with Chk1i alone, suggesting that because of the presence of 53BP1 downstream of ATM [34,35], addition of the ATMi in the combination therapy reduced the mobilization of 53BP1 to the site of DNA damage. " We have added the following text in the Discussion section (p. 13): " The combination group had more γH2AX-positive cells than the other groups, but the number of multiple 53BP1 foci–positive cells was decreased compared with the Chk1 group. It has been reported that ATM regulates 53BP1 activity [35] and that 53BP1 foci are decreased by ATM inhibitors [34]. Therefore, in combination therapy, the accumulation of 53BP1 into sites of DNA damage by the Chk1i can be suppressed by the ATMi. Thus, the inhibitory effect of ATMis on the activity of 53BP1 involved in DNA repair processes, such as homologous recombination, may prevent the repair of DNA damage caused by the Chk1i and help efficiently induce cell death. ". We have also added the following references (p. 18): “34. Isono, M.; Niimi, A.; Oike, T.; Hagiwara, Y.;, Sato, H.; Sekine, R.; Yoshida, Y.;, Isobe, S.; Obuse, C.; Nishi, R.; et al. BRCA1 di-rects the repair pathway to homologous recombination by promoting 53BP1 dephosphorylation. Cell rep. 2017, 18, 520–532. 35. Sakasai, R.; Teraoka, H.; Takagi, M.; Tibbetts, R.S. Transcription-dependent activation of ataxia telangiectasia mutated pre-vents DNA-dependent protein kinase-mediated cell death in response to topoisomerase I poison. J. Biol. Chem. 2010, 285, 15201–15208.". Your question has helped us greatly improve our manuscript.
- The data to evaluate impact of combination treatment on tumor volume seems over exaggerated especially given that the data fig 5B does not suggest a significant difference single agent treatment and combination treatment. So to conclude that the combination treatment demonstrates significant potency may not be the correct interpretation.
We agree completely with your comment. We were also concerned about the lack of promising results in vivo. In this study, only a limited number of drug concentrations and routes of administration were tested. Therefore, we intend to identify more-effective ATM and Chk1 inhibitors, routes of administration, and drug concentrations in more detail in our next study. We have therefore added the following text as a Limitation: " There were some limitations to this study. First, the antitumor effect in the in vivo model of the combined treatment was limited. Given that in vitro inhibition of Chk1 and ATM produced significant antitumor effects, it will be necessary to determine the optimal dose and route of administration that will produce the combined effect or use a different inhibitor that has superior pharmacokinetics. Preclinical studies using mice will be necessary in the future to determine whether the combination therapy has potential as a new anticancer therapy." Your comment has helped us greatly improve our manuscript.
Thank you again for providing your valuable comments regarding our manuscript. We hope that the revised manuscript is now suitable for publication.
